# Hydrological Characteristics Change of Hala Lake and Its Response to Climate Change, 1987–2018

Zhiyong Jiang [1], Jianru Wang [1], Xiaobin Cai [2,*], Junli Zhao [3], Huawei Zhang [4], Yi Zhang [5] and Chongshan Wang [1]

1 College of Earth and Environmental Sciences, Lanzhou University, Lanzhou 730000, China; zhyjiang@lzu.edu.cn (Z.J.); wangjr15@lzu.edu.cn (J.W.); wangchsh19@lzu.edu.cn (C.W.)
2 Innovation Academy for Precision Measurement Science and Technology, Chinese Academy of Sciences, Wuhan 430077, China
3 Key Laboratory of Geological Survey and Evaluation of Ministry of Education, China University of Geosciences (Wuhan), Wuhan 430074, China; zhaojunli@cug.edu.cn
4 Faculty of Geographical Sciences, Yunnan Normal University, Kunming 650000, China; hwzhang@ynnu.edu.cn
5 Technology and Innovation Centre for Environmental Geology and Geohazards Prevention, School of Earth Sciences, Lanzhou University, Lanzhou 730000, China; zhangyigeo@lzu.edu.cn
* Correspondence: xbcai@whigg.ac.cn

**Abstract:** Lakes on the Tibetan Plateau (TP) are an indicator of global climate change. The study on the factors driving lake change on the TP can help us understand its response to climate change. In this study, Landsat and ICESat data were used to obtain the variations of area, water level, and storage of Hala Lake and the area of glaciers in the Hala Lake Basin during 1987–2018. Combined with meteorological data, climate change trends and the factors driving Hala Lake change in the last 30 years were analyzed. The contribution of glacier mass loss to lake recharge was estimated by the water balance of Hala Lake. The results showed that Hala Lake has experienced three stages: slight expansion (1987–1994), shrinkage (1995–2001) and rapid expansion (2002–2018) during the study period. The rate of glacial melting continued to decline during the study period. Precipitation was the main factor that drove the hydrological characteristic changes in Hala Lake. The step change points of annual precipitation and temperature occurred in 2001, almost the same time that Hala Lake began expanding rapidly.

**Keywords:** Hala Lake changes; driving factors; glaciers change; climate change

## 1. Introduction

The Tibetan Plateau (TP), the highest and largest plateau in the world, is an amplifier of climate change. From 1982 to 2012, the average temperature rise on the TP was twice the global average [1]. There are many glaciers and lakes on the TP, which is called the Water Tower of Asia [2]. Lakes on the TP are highly sensitive to global changes due to their high altitude and low impact from human activities [3,4]. They are not only affected by precipitation and evaporation, but also by glacier retreat and permafrost degradation caused by temperature rise [5]. Therefore, it is necessary to understand lake dynamics and their relationship with climate change on the TP.

At present, the research on lakes on the TP focuses on examining the changes in lake area [6], level [7–10] and storage [11–13] based on field measurements and remote sensing data. In recent decades, lakes on the TP have generally expanded [12–15], except for lakes located in the southeastern TP [16–19]. The number and total area of lakes on the TP have also increased [20]. Lakes on the TP expanded faster than at other times during 2003–2009 [8,11].

The cause of the expansion of the lakes on the TP is a matter of debate. Zhu and Wu's research showed that the main reason for the expansion of Nam Co Lake was increased

glacier meltwater caused by rising temperatures during 1992–2004 [21]. Glacial meltwater contributes about half of the increased water storage in glacier-fed lakes in the Tanggula Mountains [22] and the northwestern part of the TP [23]. Li et al. found that the lakes in the continuous permafrost region expanded, while the glacier-fed lakes either remained stable or shrank in the isolated permafrost region; they believed that the permafrost degradation was greater than the impact of glacial melting on the lakes [14]. There was some research showing that increased precipitation on the TP is the main reason for the lakes' expansion in recent decades [10,16,24,25]. The results simulated by hydrological models also indicate that the expansion of Selin co Lake [26] and lakes in the central and southern TP [27] is caused by the increase in precipitation. Evaporation decreased [28] with dropping wind speeds on the TP [29,30]. The 4% increase in the water level of Nam Co Lake was caused by the reduction of lake evaporation during the period 1998–2008 [31].

Climate patterns over the TP are influenced by westerlies, Indian monsoon and East Asian monsoon [32–34]. Hala Lake, which is fed by glaciers and in an area with almost no human activity, is an ideal area to study the response of lakes to climate change because it is located in the transition region of the East Asian monsoon, Indian monsoon and the westerlies and is very sensitive to climate change [35].

In this study, remote sensing data was used to extract the area and water level of Hala Lake and the glacier area of Hala Lake Basin from 1987 to 2018. Changes in Hala Lake's water storage were calculated from the area and the water level. Combined with meteorological data, the factors driving change at Hala Lake were analyzed to understand the impact of climate change on the lake during 1987–2018. The change in the area of Hala Lake in 31 recent years can be divided into three stages: slight expansion (1987–1994), shrinkage (1995–2001) and rapid expansion (2002–2018). Therefore, in this study, we analyzed the driving factors of hydrological characteristics change during these three phases.

## 2. Materials and Methods

### 2.1. Study Area

Hala Lake Basin is located in the Qilian Mountains in the north of the TP, with the Basin area of 4793 km$^2$ (Figure 1). Hala Lake is a closed saline lake with an altitude of about 4078 m. The average depth and maximum depth of Hala Lake is 27.14 m and 65 m, respectively. The water storage is about $161 \times 10^8$ m$^3$. There are more than 40 short rivers with small flows and 118 glaciers that feed into Hala Lake. The climate in the Hala Lake Basin is dry and cold.

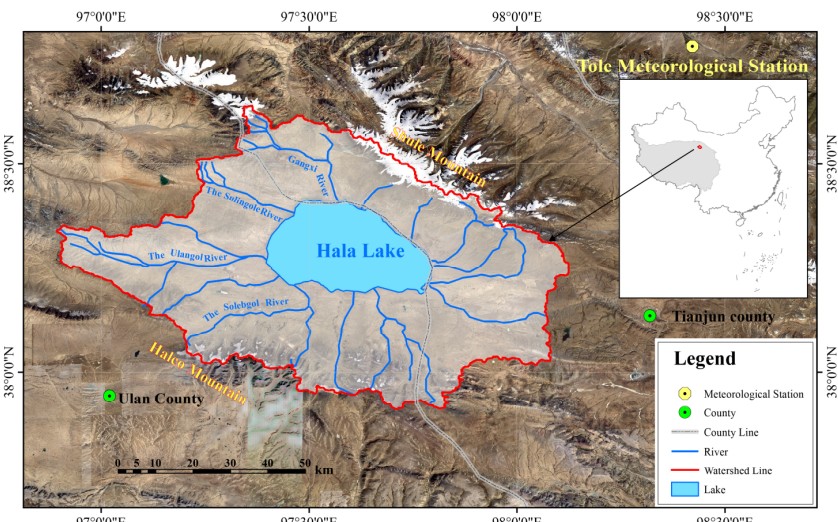

**Figure 1.** Map of the Hala Lake Basin.

### 2.2. Estimating Lake Water Storage Change and the Uncertainty

#### 2.2.1. Lake Water Storage Change Estimate

In this study, Landsat surface reflection data in Google Earth Engine (GEE) were used to extract the area of Hala Lake during 1987–2018. Only cloud-free images acquired from August to September every year were selected to avoid cloud pollution and exclude periods of lake freezing. The normalized water index algorithm(NDWI) [36] which has the highest accuracy in extracting the area of lakes on the TP [37] was used to determine the extent of Hala Lake. All the Landsat images were processed using cloud computing technology based on the GEE platform. The interannual variation of water area was estimated using the average lake area during August to September each year.

The ICESat1 data (https://search.earthdata.nasa.gov, accessed 18 December 2018) and ICESat2 data (https://nsidc.org/data/, accessed 17 June 2021) was used to extract the water level of Hala Lake from 2003 to 2009 and 2018 because elevation estimates from ICESat1 and ICESat2 are much more precise compared to those from radar altimetry [38–40]. The water level data was converted from ellipsoidal heights of WGS84 to orthometric heights of EGM2008. The level-surface area relationship curve (LRC) method correlated the water surface area with the water level data to fit the water level-area curve [41]. The LRC could be used to estimate the water level with area as the input. In this study, the LRC was constructed by the area and level of Hala Lake obtained from Landsat and ICESat data which acquired time closest:

$$H_l = 0.0957 \times A_l + 4021.3031 \left( R^2 = 0.99 \right) \tag{1}$$

where $H_l$ is the water level (m) and $A_l$ is the water area (km$^2$).

The water levels, which cannot be extracted from ICESat data, can be estimated by LRC (Figure 2) according to corresponding water areas. The following formula was used to estimate the change of water storage of Hala Lake between different years $i$ and $j$:

$$dV_{l_{i,j}} = dH_{l_{i,j}} \times \left( A_{l_i} + \sqrt{A_{l_i} A_{l_j}} + A_{l_j} \right)/3 \tag{2}$$

where $A_{l_i}$ (m$^2$) and $A_{l_j}$ (m$^2$) are the lake water areas in the year $i$ and $j$ and $dH_{l_{i,j}}$ (m) and $dV_{l_{i,j}}$ (m$^3$) are the changes in lake water levels and storage separately between year $i$ and $j$.

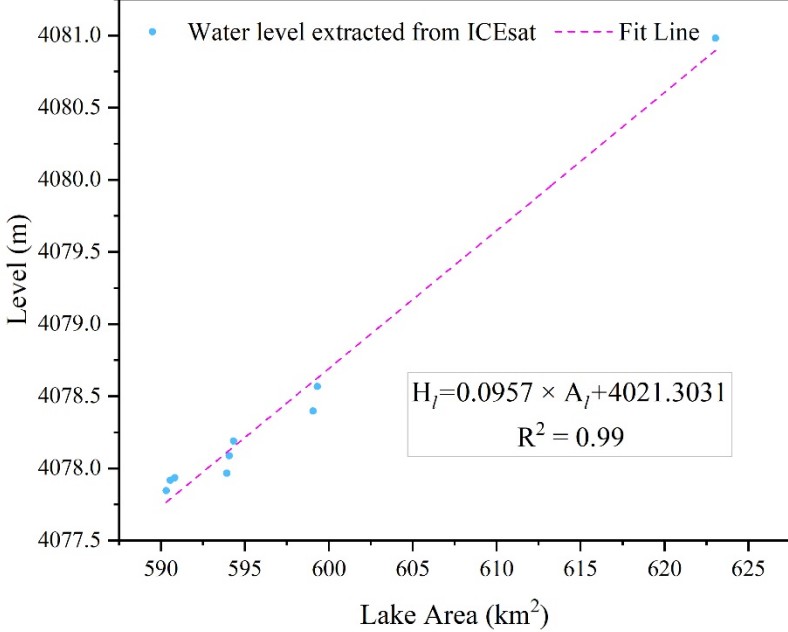

**Figure 2.** Relationship between the water level of Hala Lake and its lake area.

2.2.2. Uncertainty of Lake Water Storage Change

To obtain the LRC curve, two sets of water level-area data are sufficient, and other sets of observations are redundant. The mean square error $\sigma_{wl}$ of water level estimated by LRC can be calculated by the equation below:

$$\sigma_{wl} = \sqrt{\frac{V^T V}{r}} \tag{3}$$

where $V$ is an error vector of water levels used to construct LRC, and $r$ is the number of redundant observations. In this study, the $\sigma_{wl}$ is $\pm 0.13$ m. The median error of the difference between two water levels can be estimated using the following equation:

$$\sigma_{dH_{l_{i,j}}} = \sqrt{2}\sigma_{wl} \tag{4}$$

The difference between the area of Hala Lake abstracted by visual interpretation of a Sentinel-2 image (13 July 2018) and the result of a Landsat8 image (17 July 2018) abstracted by normalized water index algorithm is 2.59 km². Assuming that the difference above is the median error of the lake area, the relative error is only 0.4%. The median error of Hala Lake area $A_{l_i}$ in the $i$ can be calculated by a formula as follows:

$$\sigma_{A_{l_i}} = 0.4\% \times A_{l_i} \tag{5}$$

Based on the nonlinear error propagation law and Equation (2), the median error of $dV_{l_{i,j}}$ can be calculated by the following formula:

$$\sigma_{dV_{l_{i,j}}} = \frac{1}{3}\sqrt{\left(A_{l_i} + \sqrt{A_{l_i}A_{l_j}} + A_{l_j}\right)\sigma^2_{dH_{l_{i,j}}} + \left(dH_{l_{i,j}} + \frac{dH_{l_{i,j}}A_{l_j}}{2\sqrt{A_{l_i}A_{l_j}}}\right)^2 \sigma^2_{A_{l_i}} + \left(dH_{l_{i,j}} + \frac{dH_{l_{i,j}}A_{l_j}}{2\sqrt{A_{l_i}A_{l_j}}}\right)^2 \sigma^2_{A_{l_j}}} \tag{6}$$

The uncertainty of lake water storage change is given in 1 sigma confidence level. The lake level estimation is the primary source of error of the lake water storage change estimation according to Equation (6). The $\sigma_{dV_{l_{1987,1994}}}$ is larger than $dV_{l_{1987,1994}}$ because Hala Lake was relatively stable during 1987–1994. The relative error of $dV_{l_{1995,2001}}$ and $dV_{l_{2002,2018}}$ as percentages were estimated to be 18.9% and 5%, respectively.

*2.3. Estimating Glacier Mass Loss and the Uncertainty*

2.3.1. Glacier Area Extraction and Glacier Mass Loss Estimation

The seasonal snowfall may occur in all seasons except summer in the Hala Lake Basin. To avoid seasonal snowfall, Landsat images taken on cloudless days from July to September each year were selected to map the glacier extents using thresholded ratio images RED/SWIR(TM3/TM5) [42–44] in this study. Holes on the glacier map smaller than 0.05 km² which were probably caused by rock debris were repaired by a morphological reconstruction operation [45]. Only glaciers larger than 0.05 km² were mapped because a smaller area criterion would likely include many snow patches [46].

Landsat images obtained in 1990, 1995, 1999, 2002, 2006, 2010, 2013 and 2018 meet the requirements above and can be used to extract glaciers in the Hala Lake Basin. The glacier area of other years during the study period were obtained by interpolation. There was no data about the glaciers during 1987–1990 in the Hala Lake Basin; the annual change in the glacial area from 1987 to 1990 was thought to be the same as during 1990–1995.

The glacier volume–area relation can be used to estimate glacier volume change by the change of glacier area [47–50]. In this study, an empirical formula of glacier volume–area established by Liu [51] was used to estimate the glacier volume in the Hala Lake Basin:

$$V_G = 0.04 \times A_G^{1.35} \tag{7}$$

where $V_G$ and $A_G$ are glacier volume (km$^3$) and area (km$^2$), respectively. The glacier volume change can be estimated by the following equation:

$$\Delta V_{G_{i,j}} = V_{G_i} - V_{G_j} \tag{8}$$

where $V_{G_i}$ and $V_{G_j}$ are glacier volume in different years $i$ and $j$. In this study, we suppose an average glacier density of 850 kg·m$^{-3}$ [52], and glacier mass loss $Gl$ was fully converted into lake water when we estimated the impact of glaciers on Hala Lake. The lake water converted by glacier mass loss $Gl_{i,j}$ during $i - j$ years can be calculated by the following equation:

$$Gl_{i,j} = 0.85 \times \Delta V_{G_{i,j}} \tag{9}$$

2.3.2. Uncertainty of Glacier Mass Loss

The glacier area error can be estimated by the following equation [53]:

$$\varepsilon = NA_{hp} \tag{10}$$

where $\varepsilon$ (m$^2$) is the error of glacier area, $N$ is the number of pixels on the edge of the glacier and $A_{hp}$ (m$^2$) is the area of the ground corresponding to half a pixel. Assuming that the $\varepsilon$ is the median error of glacier area $\sigma_{A_G}$, the relative error $r_{G_i}$ can be calculated by the ratio of the median error $\sigma_{A_{G_i}}$ to the glacier area $A_{G_i}$ (m$^2$) in the $i$ year. In this study, only the relative error of 1990, 1995, 1999, 2002, 2006, 2010, 2013 and 2018 can be obtained. The relative errors of the other years during the study period are replaced by the average of the relative errors (4.45%) of the eight years. The median error of glacier area $A_{G_i}$ can be calculated by a formula as follows:

$$\sigma_{A_{G_i}} = r_{G_i} A_{G_i} \tag{11}$$

The equation that follows can estimate the median error of glacier volume $\sigma_{V_G}$:

$$\sigma_{V_G} = 0.054 \sigma_{A_G}{}^{0.35} \tag{12}$$

The median error of glacier volume change between different years $i$ and $j$ can be estimated by an equation as follows:

$$\sigma_{\Delta V_{G_{i,j}}} = \sqrt{\sigma_{V_{G_i}}^2 + \sigma_{V_{G_j}}^2} \tag{13}$$

The median error of glacier mass loss $\sigma_{Gl_{i,j}}$ is calculated based on $\sigma_{\Delta V_{G_{i,j}}}$:

$$\sigma_{Gl_{i,j}} = 0.85 \sigma_{\Delta V_{G_{i,j}}} \tag{14}$$

The uncertainty of glacier mass loss is also given in 1 sigma confidence level. The glacier area estimation is the primary source of error of the glacier mass loss estimation in this study. The relative error of $Gl_{1987,1994}$, $Gl_{1995,2001}$ and $Gl_{2002,2018}$ as percentages were estimated to be 5.72%, 11.88% and 5.43%, respectively.

*2.4. Water Balance*

Hala Lake Basin is an endorheic Basin, and its water budget can be expressed by the following equation:

$$dV = R + P_w - E_w \tag{15}$$

where $dV$ is the storage change of Hala Lake. Hala Lake is supplied mainly by runoff $R$ and precipitation on the lake surface $P_w$. Runoff $R$ includes groundwater exchange and surface runoff caused by land surface precipitation, permafrost thawing and glacial meltwater. The

water level of Hala Lake is affected by lake surface evaporation $E_w$. $P_w$ and $E_w$ can be calculated as the following equation:

$$P_w = Precipitation \times A_{lake} \tag{16}$$

$$E_w = E \times K \times A_{lake} \tag{17}$$

where $A_{lake}$ is the area of Hala Lake. The datasets of monthly $0.5^° \times 0.5^°$ grid-based precipitation and temperature over China, which were generated by spatial interpolation based on data of 2472 meteorological stations in China, provided by the China Meteorological Data Service Centre (https://data.cma.cn), were used to estimate the precipitation and temperature in the Hala Lake Basin. In this study, the pan evaporation $E$ (mm) from the nearest meteorological station (Tole Meteorological Station (Figure 1)) to the Hala Lake Basin were used to estimate lake surface evaporation, and $K$ is a conversion coefficient between the evaporation pan and the evaporation of the lake surface. Two kinds of evaporation pans of ∅20 and E-601B were used in the Tole Meteorological Station, with conversion coefficients of $K_{∅20} = 0.60$ and $K_{E-601B} = 0.68$, respectively [54,55].

### 2.5. Statistical Methods

The method of the non-parametric Mann–Kendall statistical test [56–58] can be used to extract step change points [37,59,60] and trends [61,62] in time series of climate parameters. For a given time series $x_1, x_2, \cdots x_n$:

$$m_i = \sum_{j=1}^{i} f(x_i - x_j) \ (1 \le i \le n) \tag{18}$$

where $f(x)$ can be defined as the following:

$$f(x) = \begin{cases} 1, & x > 0 \\ 0, & x \le 0 \end{cases} \tag{19}$$

Under the null hypothesis (no step change points), The normal distribution statistic $t_k$ can be described as follows:

$$t_k = \sum_{i=1}^{k} m_i \ (1 \le k \le n) \tag{20}$$

The mean $E(t_k)$ and variance $D(t_k)$ of $t_k$ can be defined as follows:

$$\begin{cases} E(t_k) = k(k-1)/4 \\ D(t_k) = k(k-1)(2k+5)/72 \end{cases} \tag{21}$$

$UF_k$ is the normalized variable statistic of $t_k$, and the formula is as follows:

$$UF_k = (t_k - E(t_k))/\sqrt{D(t_k)} \tag{22}$$

All $UF_k (1 \le k \le n)$ can form a curve $UF$. The time series $x_1, x_2, \cdots x_n$ has an increasing trend when $UF_k > 0$; otherwise, it has a decreasing trend. The curve $UB$ can be obtained by the above method to process the opposite time series $x_n, x_{n-1}, \cdots x_1$. The intersection of curve $UF$ and $UB$ represents the time of the step change point.

In addition, linear regression analysis and Poisson correlation analysis were used to analyze the trend of time series data and the relationship between Hala Lake water storage change and climate factors in this study.

## 3. Results

### 3.1. Changes in Area, Water Level and Storage of Hala Lake

As shown in Figure 3, the changes in the shoreline were mainly distributed in the north, southwest and southeast of the lake. The average annual area of Hala Lake was

$593.47 \pm 1.8$ km$^2$. From 1995 to 2001, the area of Hala Lake decreased continuously with a change rate of $-1.42$ km$^2 \cdot$y$^{-1}$ and reached the minimum area of 585.482 km$^2$ in 2001 (Figure 4). The area of Hala Lake expanded rapidly with a rate of 2.21 km$^2 \cdot$y$^{-1}$ from 2002 to 2018 and reached 623.031 km$^2$ in 2018.

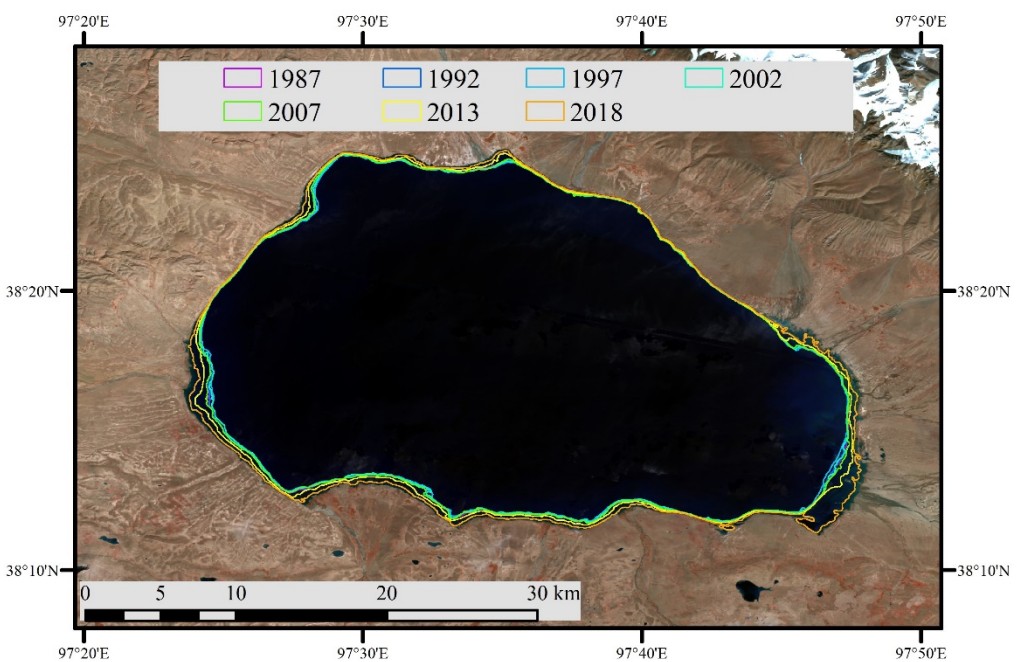

**Figure 3.** The changes in the shorelines of Hala Lake during 1987–2018.

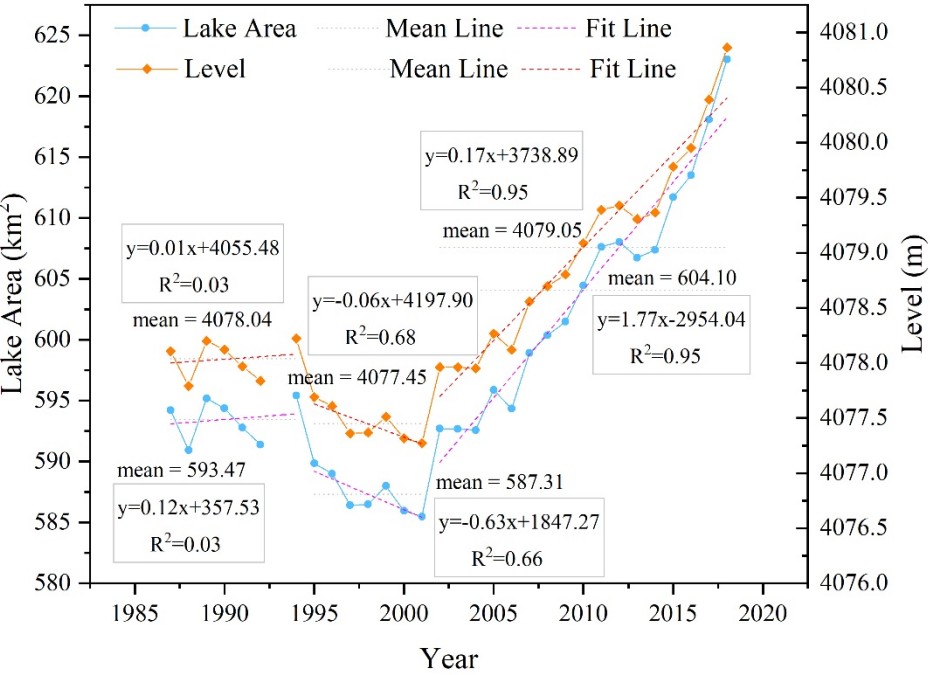

**Figure 4.** The water area and level of Hala Lake during 1987–2018.

The water level of Hala Lake increased slightly during 1987–1994 (Figure 4). The water level of Hala Lake was $4078.1 \pm 0.13$ m (Figure 4). The water level of Hala Lake decreased 0.71 m from 1995 to 2001 and reached the lowest level of 4077.272 m in 2001. The water level of Hala Lake rose by 3.59 m with a rate of 0.21 m$\cdot$y$^{-1}$ during 2002–2018 and reached a maximum of 4080.862 m in 2018 (Figure 4).

The average annual change of water storage of Hala Lake was $0.14 \times 10^8$ m$^3 \cdot$y$^{-1}$ during 1987–1994 (Figure 5). The storage of Hala Lake decreased $3.87 \times 10^8$ m$^3$ from 1995 to 2001 and increased by $21.69 \times 10^8$ m$^3$, with an average annual increase rate of about $1.28 \times 10^8$ m$^3 \cdot$y$^{-1}$, from 2002 to 2018 (Figure 5).

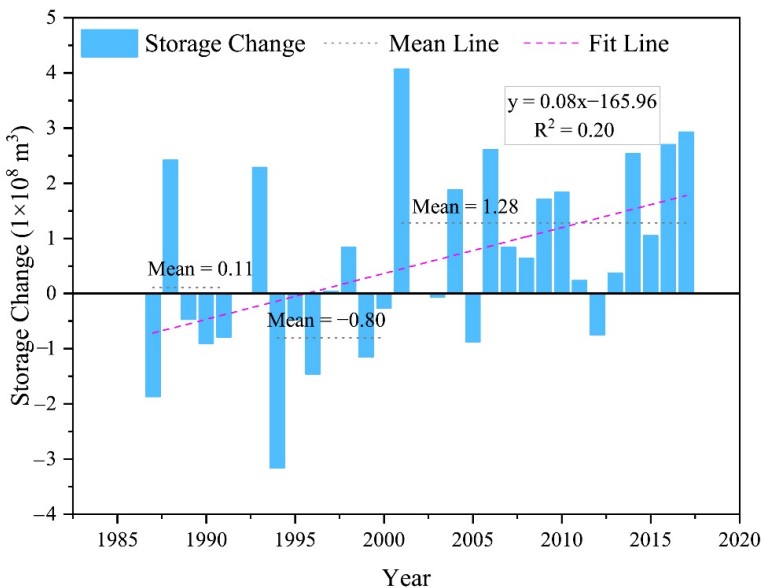

**Figure 5.** The storage change of Hala Lake during 1987–2018.

### 3.2. Glacier Area Change and Mass Loss in Hala Lake Basin

From 1987 to 2018, the glacier area of Hala Lake Basin declined (Figure 6). During 1987–1994, the glacier in Hala Lake Basin retreated about 7.07 km$^2$. The annual glacial area decreased by 1.01 km$^2 \cdot$y$^{-1}$, and annual glacial mass loss was about $2.23 \times 10^8$ m$^3 \cdot$y$^{-1}$. The glacier area shrank at a rate of 0.48 km$^2 \cdot$y$^{-1}$, and glacial mass loss was about $1.05 \times 10^8$ m$^3 \cdot$y$^{-1}$ during 1995–2001. Between 2002 and 2018, the glacier area decreased from 81.3 km$^2$ to 73.84 km$^2$. The annual glacial area declined 0.44 km$^2 \cdot$y$^{-1}$ and annual material loss was about $0.93 \times 10^8$ m$^3 \cdot$y$^{-1}$.

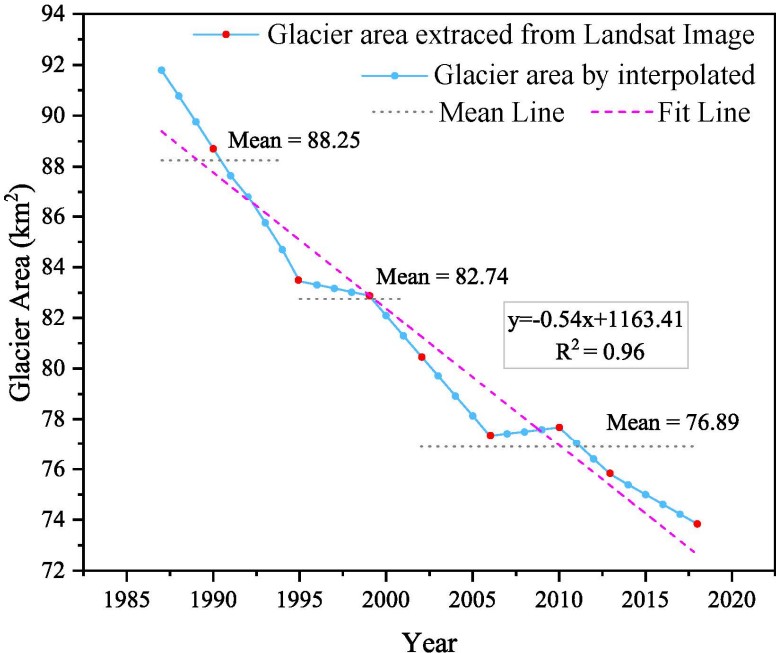

**Figure 6.** The area of glaciers in the Hala Lake Basin from 1997 to 2018.

### 3.3. Changes in Climatic Parameters

From 1987 to 2018, precipitation in the Hala Lake Basin increased with fluctuations. The average annual precipitation from 1987 to 1994 was about 343 mm. From 1995 to 2001, which was the driest during the study period, the average annual precipitation decreased 29.2 mm compared to 1987 to 1994 (Figure 7a). As illustrated in Figure 8a, the Mann–Kendall step change point test showed that the change point of annual precipitation in the Hala Lake Basin occurred in 2001. The annual average precipitation increased by 106.9 mm compared with before 2001 (Figure 7a). The wetting trend is associated with simultaneously weakened westerlies over the TP [63].

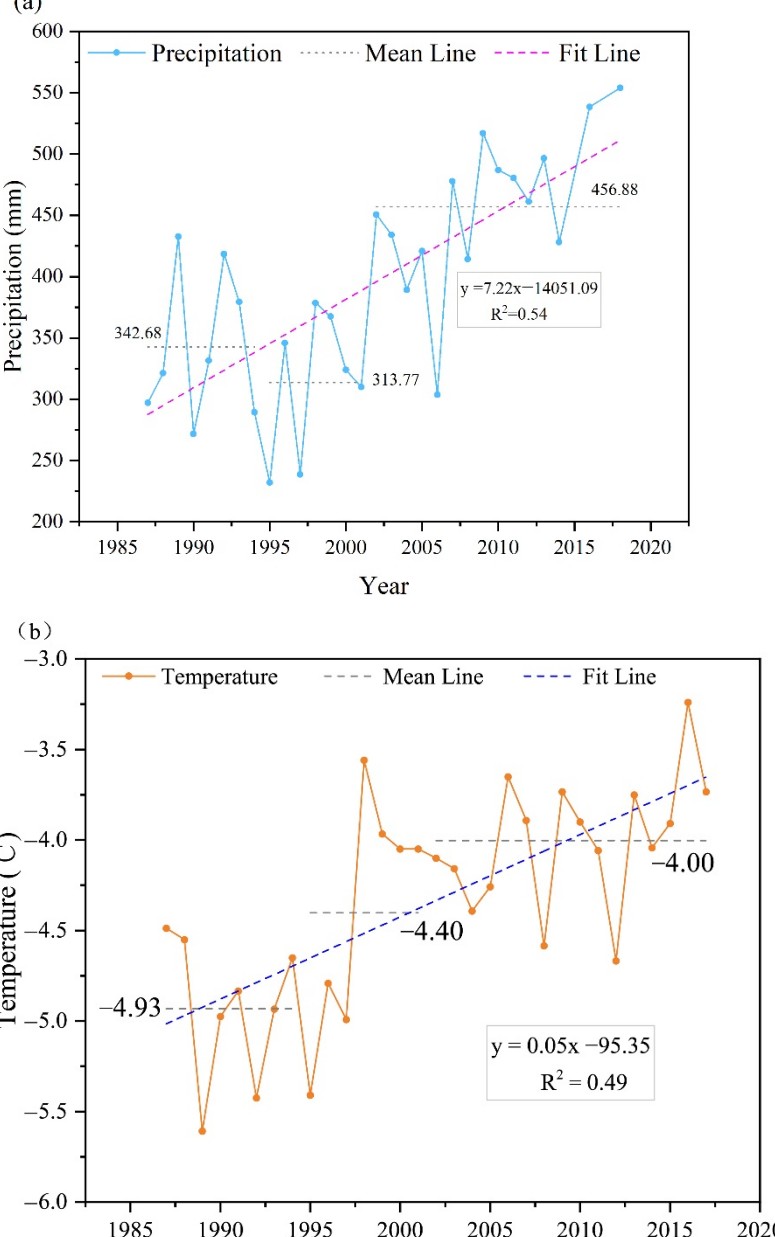

**Figure 7.** *Cont.*

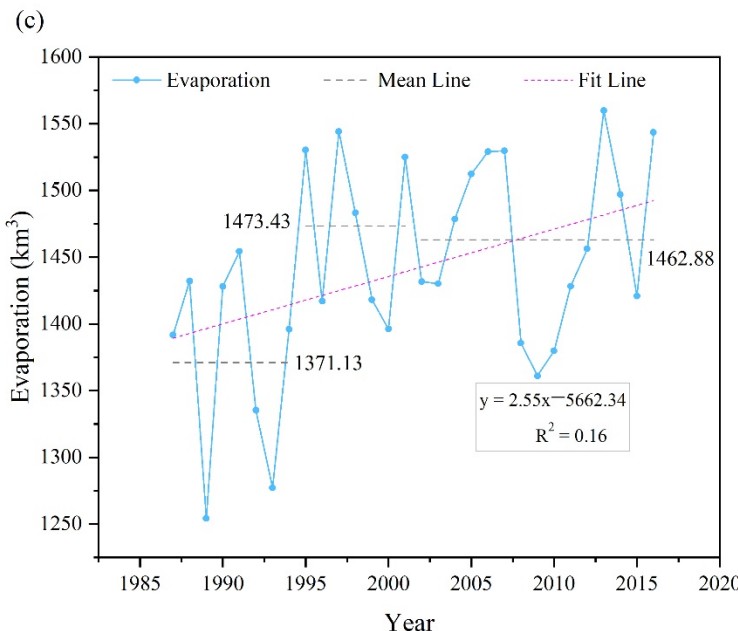

**Figure 7.** The changes in precipitation, temperature and evaporation of the Hala Lake Basin from 1987 to 2018.

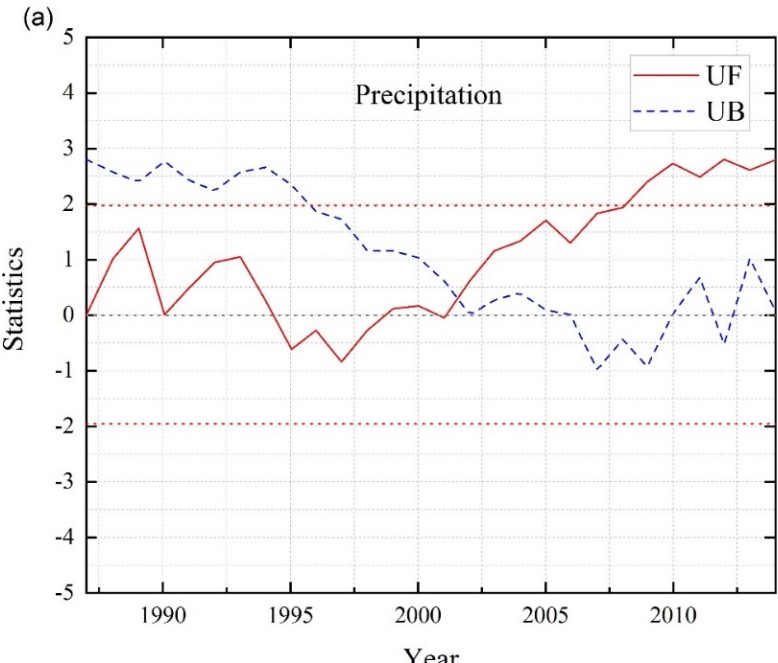

**Figure 8.** *Cont.*

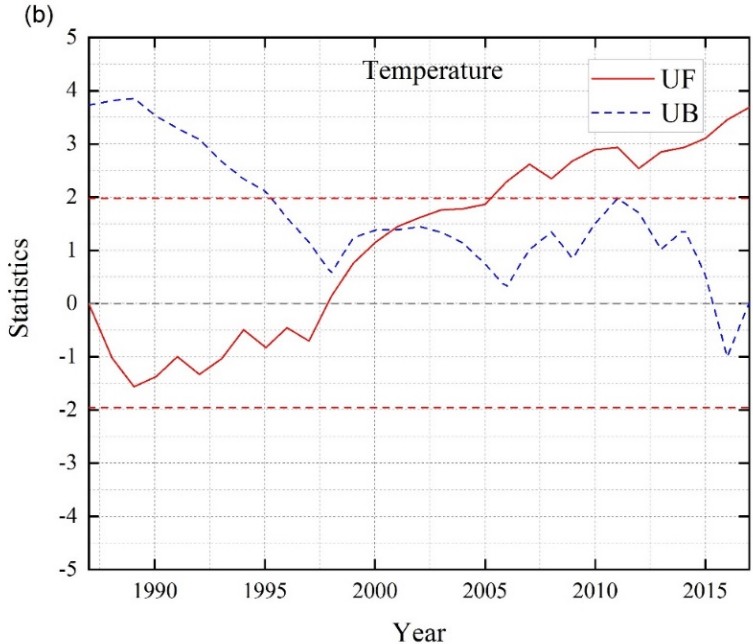

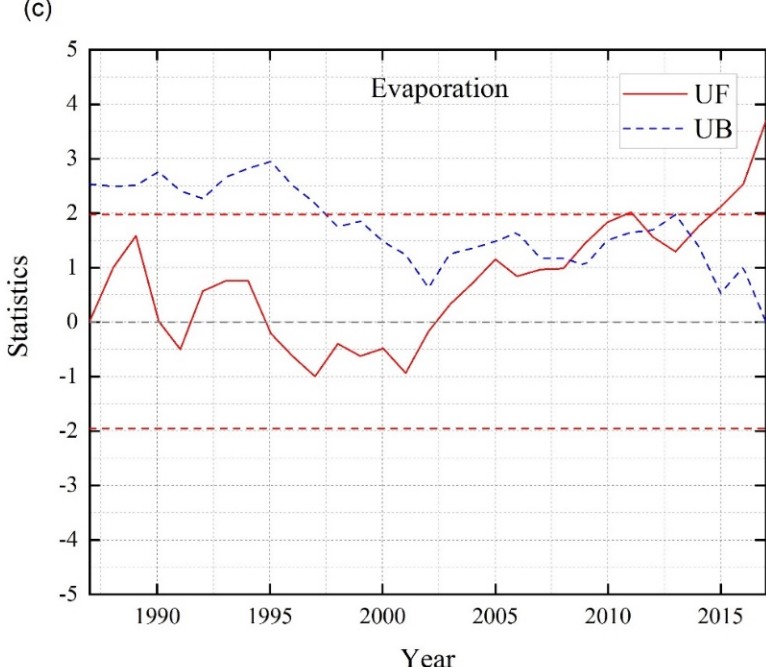

**Figure 8.** Trend test and abrupt change point detection of temperature, precipitation and evaporation in the Hala Lake watershed; the red dotted lines are the thresholds at a significant level of 95%; statistics are the values of UF and UB.

The temperature of Hala Lake Basin was on the rise during the research period. During 1987–1994, the average annual temperature was −4.97 °C. The average annual temperature of 1995–2001 was 0.57 °C higher than that from 1987 to 1994 (Figure 7b). The temperature change point in the Hala Lake Basin occurred in 2001 (Figure 8b). The average annual temperature reached −4.0 °C after 2001 (Figure 7b).

The evaporation in the Hala Lake Basin also increased with fluctuations. The annual average evaporation was 1371.5 mm during 1987–1994 and increased to 1473 mm from 1995 to 2001. During 2002–2018, the average annual evaporation was 1462 mm which was 90.5mm higher than the first stage but slightly lower than that from 1995 to 2001 (Figure 7c).

The results of the Mann–Kendall test cannot indicate the change point of evaporation because the curves $UF$ and $UB$ intersect at three points (Figure 8c).

*3.4. Water Balance of Hala Lake*

The water supply of Hala Lake is mainly through runoff $R$ and lake surface precipitation $P_w$, and water loss is mainly through lake surface evaporation $E_w$. The difference between supply and loss is the water storage change $dV$.

As shown in Table 1, the average annual runoff and lake surface precipitation were about $3.00 \times 10^8$ m$^3 \cdot$y$^{-1}$ and $2.03 \times 10^8$ m$^3 \cdot$y$^{-1}$, respectively, and the average annual lake surface evaporation was $4.92 \times 10^8$ m$^3 \cdot$y$^{-1}$ from 1987 to 1994. At the same time, the average annual water storage change was $0.11 \times 10^8$ m$^3 \cdot$y$^{-1}$. The area of Hala Lake experienced a slight increase during this period.

**Table 1.** The Hala Lake water balance from 1987 to 2018.

| Water Balance Components | | 1987–1994 Annual Average $(1 \times 10^8$ m$^3$ y$^{-1})$ | 1995–2001 Annual Average $(1 \times 10^6$ m$^3$ y$^{-1})$ | 2002–2018 Annual Average $(1 \times 10^6$ m$^3$ y$^{-1})$ |
|---|---|---|---|---|
| Lake water supply | $R$ | 3.0 | 2.46 | 3.85 |
| | $P_w$ | 2.03 | 1.93 | 2.76 |
| Loss of lake water | $E_W$ | 4.92 | 5.19 | 5.28 |
| $dv$ * | | +0.11 | −0.80 | +1.33 |

* "−" means a decrease in water storage; "+" indicates an increase in water storage.

Hala Lake dropped in level and area during 1995–2001. The average annual runoff and lake surface precipitation in Hala Lake decreased by $0.54 \times 10^8$ m$^3 \cdot$y$^{-1}$ and $0.1 \times 10^8$ m$^3 \cdot$y$^{-1}$ than during 1987 to 1994, respectively, while evaporation increased, and the average annual water storage changed by $-0.80 \times 10^8$ m$^3 \cdot$y$^{-1}$.

The area of Hala Lake expanded rapidly from 2002 to 2018. The average annual runoff and water surface precipitation increased to $3.85 \times 10^8$ m$^3 \cdot$y$^{-1}$ and $2.76 \times 10^8$ m$^3 \cdot$y$^{-1}$, respectively. Although lake evaporation increased with the expansion of Hala Lake, water supply was greater than loss, and the average annual water storage increased by $1.33 \times 10^8$ m$^3 \cdot$y$^{-1}$.

## 4. Discussion on Storage Change of Hala Lake

*4.1. The Driving Factors of Water Storage Change of Hala Lake*

The water storage change of Hala Lake is the result of a variety of hydrometeorological factors. There was no significant change in precipitation and temperature in the Hala Lake Basin ($p < 0.01$) during 1987–1994. The glaciers retreated at the fastest rate in this stage over the study period. The annual average glacial mass loss account for 44% of lake water supply. The slight increase of water storage in Hala Lake (Table 1) may be caused by the mass loss of glaciers and the relatively low evaporation in the Basin at that time.

The water storage change of Hala Lake was affected mainly by precipitation and temperature (Table 2) from 1995 to 2001. The glacier continued to shrink with the temperature rise in the Hala Lake Basin. However, the annual average glacial mass loss only accounted for 24% of the lake's water supply as the rate of glacial retreat declined. At the same time, precipitation was also on the decline in this stage. The lake water supply was $0.61 \times 10^8$ m$^3 \cdot$y$^{-1}$ lower than during 1987–1994. On the other hand, evaporation was the highest over the study period. The decline in precipitation and glacial mass loss with the increase in evaporation led to the shrinking of Hala Lake during this period.

**Table 2.** The Pearson correlation coefficient between the storage change of Hala Lake and climate factors.

| $dv$ | $P$ | $T$ | $E$ |
|---|---|---|---|
| 1987–2018 | 0.4 ** | 0.31 + | −0.06 |
| 1987–1994 | −0.28 | 0.04 | −0.14 |
| 1995–2001 | 0.91 ** | 0.80 * | −0.54 |
| 2002–2018 | 0.45+ | 0.07 | −0.20 |

Note: $+ \; p < 0.1$; $* \; p < 0.05$; $** \; p < 0.01$ (significant at these levels). $dv$: Storage change of Hala Lake; $P$: Precipitation; $T$: Temperature; $E$: Evaporation.

During 2002–2018, the glaciers retreated at the lowest rate of the study period. The annual average glacial mass loss accounted for only 14.2% of the water supply of Hala Lake. Remarkably, there was even a slight increase in the glacier area between 2006 and 2010. The annual average evaporation decreased slightly (Figure 8c) compared with 1995 to 2001. Less evaporation could be caused by more clouds and less solar radiation. However, the water lost to evaporation from the lake surface was higher than that of the previous stages (Table 1). Hala Lake was still in an expanding state which indicated that the increase of precipitation offset the effects of lake surface evaporation increase and glacial mass loss decrease and was the main reason for the expansion of Hala Lake.

*4.2. Compared with Previous Studies*

Comparing previous publications with this study, it can be seen that the changes of lakes and climate on the TP have spatial–temporal differences. In the southeast of the TP, Yamzho Yumco [64] and Paiku co [17] have decreased in water area and level in recent decades. In the central TP, Nam co Lake began to expand in 1971, and expansion accelerated after 1992 [21]. Co Nag, Daru Co, Dung Co, Pung Co, Co Ngoin, Bam Co and Neri Punco have experienced an obvious increase since 1997. The areas of Cuona Lake and Zigetang Lake greatly expanded after 1998 [65]. The step change points in precipitation in the central TP occurred in 1997 [37]. Aksai Chin Lake and Bangdag Co, located in the cool and dry northwest of the TP, have expanded since 1997 [66]. In the northeast of the TP, Hala Lake and Qinghai Lake began to expand after 2001 and 2004, respectively [6]. The step change points of temperature and precipitation in the Hala Lake Basin occurred in 2001.

The spatial–temporal differences in changes of lakes and climate over the TP could be connected with changes in atmospheric circulation from the 1990s. In the southeast of the TP, the decrease in rainfall, which is associated with a weakening of the Southern Oscillation and relaxation of the meridional temperature gradient over the Indian Ocean [67], lead to the shrinkage of lakes. In the inner TP, the increase of summer precipitation could be linked with the AMO. The AMO has been in a positive phase (warm anomaly of the North Atlantic Ocean surface) since the mid-1990s, which has led to both a northward shift and weakening of the subtropical westerly jet stream at 200 hPa near the TP through a wave train of cyclonic and anticyclonic anomalies over Eurasia. These anomalies are characterized by an anomalous anticyclone to the east of the internal TP and an anomalous cyclone to the west of the internal TP. The former weakens the westerly winds, trapping water vapor over the inner TP, while the latter facilitates water vapor intruding from the Arabian Sea into the inner TP [63]. The poleward shift of the East Asian westerly jet stream [68] and the enhanced East Asian Summer Monsoon circulation under a warming climate [69] could be responsible for the wetting trend in the north (including the northwest and the northeast) of the TP.

Some studies have revealed that glacier melting has been accelerated in the past decade over the TP [70–72]. However, the rate of glacial retreat in the Hala Lake Basin has slowed over the study period. The contribution of glacial mass loss on lake expansion on the TP has also showed spatial–temporal variability. The study of Lei et al. [73] showed that glacier mass loss accounted for 11.7%, 28.7% and 11.4% of the total lake level rise of Siling Co, Nam Co and Pung Co, respectively. Ke et al. [72] revealed that the massive

lake water increase was essentially not from the mass loss of glaciers which represents only about $4.7 \pm 8.8\%$ of the lake water change. Brun et al. [74] demonstrated that glacier mass loss did have a limited contribution to the lake volume increase ($19 \pm 21\%$ for the whole Tibetan plateau) from 1994 to 2015. The results of Wu [75] showed that the glacier mass loss in the Hala Lake Basin accounted for 39.65% of the increase of water storage in Hala Lake during 2000–2016. The contribution of glacier mass loss to the Hala Lake expansion was overestimated in the study of Wu because he did not consider water surface evaporation, which is the main consumption mode of inland lakes.

## 5. Conclusions

In this study, changes in Hala Lake, glaciers and climate change in the Hala Lake Basin were investigated. The driving factors of the changing dynamics of Hala Lake were analyzed.

During 1987–2018, Hala Lake has undergone three stages of slight expansion, shrinkage and rapid expansion. During 1987–1994, the area of Hala Lake was relatively stable, with an average area of 593.47 $km^2$ and an average water level of 4078.1 m. The water level and storage of Hala Lake decreased by 0.95 m and $5.61 \times 10^8$ $m^3$ from 1995 to 2001 respectively. The area of Hala Lake shrank to 585.482 $km^2$ in 2001, which was the smallest over the study period. Hala Lake expanded rapidly from 2002 to 2018. By 2018, the area had expanded to 623.10 $km^2$. The water level and storage increased by 3.59 m and $21.69 \times 10^8$ $m^3$ during 2002–2018.

The rate of glacier melting continued to decline during the study period. The annual average glacial mass loss due to rising temperatures accounted for 44%, 24% and 14.2% of lake recharge during 1987–1994, 1995–2001, 2002–2018, respectively.

Precipitation is the main factor in the rapid expansion of Hala Lake during 2002–2018. This study also found that the time of Hala Lake's rapid expansion was closely associated with the step change points of annual precipitation and temperature based on the Mann–Kendall test.

From 2011 to 2100, the overall change in regional average annual precipitation is not evident, and air temperatures increase monotonously in the Qilian Mountains [76]. An increase in meltwater from glaciers caused by higher temperatures could lead to the expansion of Hala Lake. At the same time, higher temperatures and larger areas of Hala Lake also led to more consumption of lake water. Hala Lake will be in a stable state when lake water supply and consumption balance in the future.

**Author Contributions:** Conceptualization, Z.J. and J.W.; methodology, H.Z.; software, Y.Z.; validation, J.Z.; data curation, H.Z. and C.W.; writing—original draft preparation, Z.J. and J.W.; writing—review and editing, X.C. All authors have read and agreed to the published version of the manuscript.

**Funding:** This research supported by the National Natural Science Foundation of China (Grand No.41201437; Grand No.41861015; Grand No.42171381) and the Fundamental Research Funds for the Central Universities (lzujbky-2021-ey05).

**Data Availability Statement:** Data available on request from the first author.

**Conflicts of Interest:** The authors declare no conflict of interest.

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
