# Peer review of "Hydrological Characteristics Change of Hala Lake and Its Response to Climate Change, 1987–2018"

_remotesensing, doi:10.3390/rs14122886_

Round 1

Reviewer 1 Report

This manuscript is interesting in response of lake to climate change. It is good in total. However, if the following can be addressed, it would be more improved:

  1. The lakes water storage was estimated by ICESat data. Besides the uncertainty of the storage change, it is better to describe more detailed the estimation method and procedure.
  2. In the section of “3. Results”, it is better to give a latest curve plotting of Hala Lake.
  3. It is better to give a detailed description of data sources of the evaporation, precipitation, temperature.

In the section of “4. Discussion”, it is better to predict this lake change in the future.

Author Response

The lakes water storage was estimated by ICESat data. Besides the uncertainty of the storage change, it is better to describe more detailed the estimation method and procedure.

Reply:Thanks for your suggestion. More detailed descriptions of the estimation method were added in line 100-101 and 108-109.

In the section of “3. Results”, it is better to give a latest curve plotting of Hala Lake.

Reply: Thanks for your nice suggestion. The latest curve of Hala Lake has been drawn and included in the paper (Fig. 2).

It is better to give a detailed description of data sources of the evaporation, precipitation, temperature.

Reply: Thanks for your suggestion. More detailed descriptions of the’ datasets of monthly  grid-based precipitation and temperature over China’ were added in line 202-203.

In the section of “4. Discussion”, it is better to predict this lake change in the future.

Reply: Thanks for your suggestion. The prediction of Hala Lake change in the future had been done based on the research of Liu, L.-Y et al. [1] and added in line 479-484.

Reference

  1. Liu, L.-Y.; Wang, X.-J.; Gou, X.-H.; Yang, M.-X.; Zhang, Z.-H. Projections of surface air temperature and precipitation in the 21st century in the Qilian Mountains, Northwest China, using REMO in the CORDEX. Advances in Climate Change Research 2022, 13, 344-358, doi:10.1016/j.accre.2022.03.003.

Thanks again for all the constructive comments and useful suggestions that helped us to improve the quality of our manuscript. We hope that these revisions in the manuscript and our accompanying responses will be adequate to make our manuscript suitable for possible publication. Do not hesitate to contact us if there are any other questions and suggestions regarding the revisions of this manuscript, we will continue to revise it. We shall look forward to hearing from you at your earliest convenience.  

Sincerely yours 

Corresponding authors 

Reviewer 2 Report

In the manuscript, the authors investigated the changes in Hala Lake, glaciers and climate change in Hala Lake Basin using the remote sensing data. Then the driving factors of dynamics of Hala Lake were analyzed. This scientific issue is interesting.

Some suggestions:

The method is somewhat roughness, leading the result is not convincible.  For example: the volume-area relationship is an empirical formula, which is maybe not suitable to the glaciers around Hala lake.

To put the Fig.3,Fig,4,Fig5 into one figure is better for reading.

Where does the runoff data in Table come from? 

Author Response

  1. The method is somewhat roughness, leading the result is not convincible.  For example: the volume-area relationship is an empirical formula, which is maybe not suitable to the glaciers around Hala lake.

Reply: Thanks for your suggestion. The formula, proposed by Liu et al.[1], aimed at glaciers in the western Qilian Mountains, was suitable for estimating the volume of glaciers in this study because the Hala Lake Basin belongs to the western Qilian Mountains.

  1. To put the Fig.3, Fig,4, Fig5 into one figure is better for reading.

Reply: Thanks for your nice suggestion. The attempt to put Fig.1, Fig.2, and Fig.3 into one figure failed. We put Figure 1 and Figure 2 into one figure in order to be better for reading. 

  1. Where does the runoff data in Table come from? 

Reply: Thanks for your suggestion. In this study, the groundwater exchange and surface runoff both are difficult to be estimated due to the absence of data. We have to define ‘runoff’ as ‘Runoff R includes groundwater exchange and surface runoff caused by land surface precipitation, permafrost thawing, glacial melt water’ in line 194-196. The runoff data in Table were calculated by equation.15. In this equation, storage change of Hala Lake, precipitation on lake surface and lake surface evaporation can be calculated by equation.2, equation.16 and equation.17 respectively.  

Reference

  1. Liu, S.; Sun, W.; Shen, Y.; Gang, L. Glacier changes since the Little Ice Age maximum in the western Qilian Shan, northwest China, and consequences of glacier runoff for water supply. Journal of Glaciology 2003, 49, 117-124, doi:10.3189/172756503781830926.

Reviewer 3 Report

In this study, the authors examined the lake area, level and volume changes for Hala Lake in the Tibetan Plateau. Furthermore, the lake water balance was analyzed. Overall, this study is too simple, and did not provide some new knowledge. Especially, the water balance was only qualitatively analyzed using statistical method. For example, the glacier volume-area relationship was used to estimate volume change by input of area. This is very old method, and cannot be used now as geodetic method has been widely used to quantity glacier mass balance. I can not support the publication of this study in Remote Sensing as too simple and even not reliable results presented.  

Author Response

In this study, the authors examined the lake area, level and volume changes for Hala Lake in the Tibetan Plateau. Furthermore, the lake water balance was analyzed. Overall, this study is too simple, and did not provide some new knowledge. Especially, the water balance was only qualitatively analyzed using statistical method. For example, the glacier volume-area relationship was used to estimate volume change by input of area. This is very old method, and cannot be used now as geodetic method has been widely used to quantity glacier mass balance. I can not support the publication of this study in Remote Sensing as too simple and even not reliable results presented.  

Reply:Thanks for your suggestion. Firstly, the method, which estimates glacier volume change by input of glacier area according to the glacier volume-area relationship, still was used in some recent research[1-4]. Secondly, the formula, proposed by Liu et al.[5], aimed at glaciers in the western Qilian Mountains, was suitable for estimating the volume of glaciers in this study because the Hala Lake Basin belongs to the western Qilian Mountains. Thirdly, we have to use empirical formula to estimate glacier volume in this study because there is no in situ measured data of glacier thickness which is an essential parameter for geodetic method.

conference

  1. Radić, V.; Hock, R. Regional and global volumes of glaciers derived from statistical upscaling of glacier inventory data. Journal of Geophysical Research 2010.
  2. Adhikari, S.; Marshall, S.J. Glacier volume-area relation for high-order mechanics and transient glacier states. Geophysical Research Letters 2012, 39, 132-133.
  3. Huss, M.; Farinotti, D. Distributed ice thickness and volume of all glaciers around the globe. Journal of Geophysical Research Atmospheres 2012, 117, F04010.
  4. Grinsted; A. An estimate of global glacier volume. The Cryosphere Discussions 2013, 7, 2888.
  5. Liu, S.; Sun, W.; Shen, Y.; Gang, L. Glacier changes since the Little Ice Age maximum in the western Qilian Shan, northwest China, and consequences of glacier runoff for water supply. Journal of Glaciology 2003, 49, 117-124, doi:10.3189/172756503781830926.

Reviewer 4 Report

This article is an intersting and very important research. The article tries to undertsand some fundamental questions of today, climate change and impacts. 

  1. Area/volume and depth (water level) is non linear for natural lakes but in figure 2 it is shown as linear. It can only be linear if the shape is rectangular or square.
  2. Is there any bathymetric data of this lake? This might answer the question above.
  3. The water balance of the lake (equation 15)  ignores the unknown (to me at least) losses like groundwater leakage, unless it is well established that there is no such losses. The equation (equation 15) should then look like this:  dV = R +Pw -Ew +/-losses
  4. Is there a groundwater connection between the glaciers and lake?
  5. The 3 periods (shrinking, expansion, rapid exapansion) brings out the complexity of attribution. 1995 -2000 periods is similar to 2005 - 2010 in terms of glacier area (directly meaured or extracted from landsat image) see figure 6.   When this figure is compared to figure 7a, it can be concluded that lake area is dependent on precipitation as indicated in line 26 by [26]. In any case, precipitation increases glacier melt!!!  
